# Can Perceived Exertion and Velocity Loss Serve as Indirect Indicators of Muscle Fatigue During Explosive Back Squat Exercise?

**DOI:** 10.3390/jfmk9040238

**Published:** 2024-11-16

**Authors:** Hanye Zhao, Takanori Kurokawa, Masayoshi Tajima, Zijian Liu, Junichi Okada

**Affiliations:** 1Faculty of Sport Sciences, Waseda University, Tokorozawa 359-1192, Japan; jun1okd@waseda.jp; 2Graduate School of Sport Sciences, Waseda University, Tokorozawa 359-1192, Japan; takanori1713@fuji.waseda.jp (T.K.); masayoshitajima749@gmail.com (M.T.); ryuushikenn10611@fuji.waseda.jp (Z.L.)

**Keywords:** neuromuscular fatigue, Borg scale, surface electromyography, resistance training

## Abstract

**Background:** Muscle fatigue is inevitable during resistance exercises, making its monitoring essential for maintaining athletic performance and reducing the risk of injury. Ratings of perceived exertion (RPE) and velocity loss have been reported as reliable indicators of muscle fatigue during explosive resistance exercises. However, their validity has been assessed only indirectly. This study aimed to directly examine the validity of RPE and velocity loss as markers of muscle fatigue during explosive back squat (BS) exercises. **Methods:** Seventeen trained men performed three BS tasks with varying volumes (low, medium, high) at 65% of their one-repetition maximum. RPE, spectral fatigue index (SFI), and velocity loss were measured throughout the tasks. **Results:** Significant effects were observed across conditions for overall RPE (*p* < 0.001) and velocity loss (*p* < 0.001), while no significant effect was found for SFI. RPE and SFI increased significantly as the tasks progressed (*p* < 0.001), while velocity did not significantly decrease. Significant but weak correlations were found between RPE and SFI (*r* = 0.325, *p* < 0.001) and between velocity loss and SFI (*r* = 0.224, *p* < 0.001). **Conclusions:** These findings suggest that RPE and muscle fatigue levels increase correspondingly, indicating that RPE could serve as a practical, indirect fatigue marker for explosive BS exercises. However, velocity loss may not fully reflect muscle fatigue during lower-body explosive training and should not be used as the sole indicator. Additionally, caution is warranted when applying velocity-related parameters as indirect physiological markers for resistance exercises. The significant but weak correlations between RPE, velocity loss, and SFI suggest that assessing muscle fatigue in lower-body exercises remains challenging.

## 1. Introduction

Muscle fatigue is a common phenomenon experienced in everyday life. It can occur in various contexts, such as during physical labor and nearly all sports-related activities. When performing resistance exercises, muscle fatigue is generally considered unavoidable [1]. When muscle fatigue occurs, the muscle’s ability to generate force is compromised, which negatively impacts explosive athletic performance, including peak velocity and power output [2,3]. These impairments in force-generating capacity have also been linked to an increased risk of injuries. The heightened injury risk primarily results from a breakdown in exercise technique and proper posture [4]. Furthermore, some injuries have been reported to increase the risk of developing chronic pain conditions, such as lower back pain [5,6]. Therefore, assessing muscle fatigue is crucial for achieving optimal athletic performance, ensuring safety, and reducing injury risk across all types of resistance exercise scenarios.

Muscle fatigue is influenced by various physiological and psychological responses, affecting both central and peripheral systems. As a result, it can be evaluated using a range of measurements. Maximal force output is one of the most commonly used indicators of muscle fatigue and is widely applied in clinical, physiology, and sports science research [2,7]. However, assessing maximal force typically requires specialized equipment, such as dynamometer systems (e.g., the Biodex system), which limits the availability of this method to specific research and medical settings. Surface electromyography (sEMG) is a widely used, non-invasive tool for evaluating neuromuscular function in medical, research, and exercise-related settings [8,9]. Neuromuscular fatigue can be quantified using sEMG by analyzing the power spectrum of the sEMG signal [2,9]. Over the past century, extensive research has focused on assessing muscle fatigue using sEMG, and the precision of sEMG-based assessments has reached a high level [2,10]. When using sEMG to measure muscle fatigue, physiological changes in the muscle fibers, such as increased hydrogen ion concentration and disruption of homeostasis, can impact muscle fiber conduction velocity [11]. These changes in muscle fiber conduction characteristics lead to a shift in the power spectrum toward lower frequencies [9,10,12]. Consequently, decreases in both the mean and median frequencies, which are altered by changes in power spectrum characteristics, are regarded as standard markers of muscle fatigue in sEMG-based assessments. [9,13,14].

Understanding muscle fatigue effectively and simply is essential not only for athletes and coaches, but also for the general population, such as resistance training enthusiasts. With the increasing use of sEMG in related fields, some researchers and healthcare professionals have begun incorporating sEMG in exercise and research settings [9,15,16]. However, sEMG equipment is generally expensive, making it nearly inaccessible for individual users. Even when sEMG devices are available, interpreting the data requires a high level of expertise [9]. For example, sEMG signals can become unstable during resistance exercises, requiring advanced skills in signal processing and analysis [14]. These characteristics of sEMG limit its practicality for large-scale use or use by individuals outside of related fields. Moreover, in explosive resistance training programs, timely feedback on muscle fatigue is essential for maintaining explosive athletic performance (e.g., peak power and velocity). However, fatigue-induced shifts in the power spectrum typically occur over a longer timescale, making it less effective for real-time monitoring. When the training goal is to enhance explosive athletic performance, athletes must lift weights as explosively as possible while minimizing any significant velocity decreases caused by muscle fatigue [3]. In such scenarios, sEMG-based muscle fatigue assessment has limited effectiveness, as it cannot provide the real-time feedback necessary for explosive resistance training programs.

To address the limitations in muscle fatigue assessment, several indirect muscle fatigue markers have been proposed [3,17,18]. For example, blood lactate, ammonia, and cortisol concentrations have been suggested as valid indirect indicators of muscle fatigue [19]. However, these physiological measures often involve invasive and/or chemical methods, such as blood and saliva sampling, which can impose unnecessary physical and psychological burdens on participants. Furthermore, these measures can be easily influenced by various factors specific to resistance exercises, such as changes in relative intensity (percentage of repetition maximum, %RM) and variations in rest intervals [20,21]. These characteristics of resistance exercises make it challenging to eliminate such influences. More recently, non-invasive, indirect fatigue indicators have been advocated for fatigue assessment. For example, the jump height from countermovement jumps pre- and post-exercise has been used as a simplified marker of fatigue in explosive resistance exercises [3]. However, similar to maximum force measurement, assessing jump height also requires specialized devices, such as force plates and accelerometers. Furthermore, these measures may not be suitable for intra-set muscle fatigue assessment. Consequently, muscle fatigue assessment using physiological measures is limited in its scope of application and is currently suitable only for research and clinical settings.

In the past decade, velocity-based training has become increasingly popular in explosive resistance exercise settings [22]. Previous studies have shown that velocity-based feedback has positive effects on enhancing and/or maintaining athletic performance. Consequently, over the past decade, numerous new velocity monitoring devices at affordable prices have been developed [23,24,25]. These devices typically use built-in rotary encoders or accelerometers to measure lifting velocity by tracking the time and displacement of a barbell or dumbbell during the lifting process. With the growing use of velocity-based training, velocity loss has been proposed as a valid indirect marker of muscle fatigue [3,26]. Velocity loss has been shown to significantly correlate with blood lactate and ammonia concentrations, which are physiological markers of fatigue during explosive resistance exercises [3,27]. However, applying this fatigue assessment during resistance exercises requires high-precision velocity measuring devices, making velocity-based fatigue assessment impractical for most individuals. Additionally, for practitioners with limited resistance training experience, such as beginners, enthusiasts, or inexperienced athletes, performing lifts in an explosive manner may be unsuitable. These individuals often lack sufficient muscle strength, joint stability, and proper lifting techniques [28]. Additionally, lifting explosively is not suitable for resistance training programs focused on rehabilitation and muscular hypertrophy [29]. Therefore, velocity-based fatigue assessment should be applied only to skilled lifters participating in explosive training programs.

The rating of perceived exertion (RPE) scale is a perception-based method commonly used to quantify exercise intensity in both clinical and research settings [30,31,32]. RPE was originally developed to assess the intensity of medical treatments. However, it has since been widely adopted in research and practical exercise programs due to its ease of use and reliability in reflecting physiological responses. For example, RPE has been shown to provide reliable information on physiological responses related to physical activity, such as heart rate, maximal oxygen consumption, muscle activation, and fatigue levels during aerobic exercise [33,34]. Over the past two decades, with the increasing global popularity of resistance exercise, RPE has garnered significant attention in resistance exercise-related studies. RPE has been shown to reflect several important variables in resistance training, including %1RM, muscle activation levels, exercises array, and rest interval configurations [21,32,35,36]. Due to its reliability, many practitioners have started using RPE to monitor resistance training programs. Several types of RPE scales currently exist for practical resistance training applications (e.g., the OMNI-RES scale and Borg’s scale). Most of these RPE scales have been validated and are considered reliable for monitoring physiological indices during resistance exercises [21,31,32]. Therefore, RPE may also reflect muscle fatigue responses in explosive resistance exercises. Regarding the relationship between RPE and muscle fatigue, previous studies have shown that RPE correlates with muscle fatigue levels during non-explosive resistance exercises [37,38,39]. However, the relationship between RPE and muscle fatigue in explosive resistance exercises remains unclear, as muscle fatigue assessments (e.g., sEMG) in these contexts require extensive expertise and experience [9,14]. Overall, the validity of RPE-based muscle fatigue assessments has been investigated only in specific types of resistance exercises [16,39,40]. It remains unclear whether RPE can serve as a versatile marker of muscle fatigue, including during explosive resistance exercises.

To address the challenges associated with non-invasive muscle fatigue assessment, several new mathematical sEMG-based algorithms have been developed [10,12,14]. These new methods offer greater precision in muscle fatigue assessment and may enable sEMG-based evaluations of muscle fatigue during explosive movements. Furthermore, they provide the potential to develop an RPE-based muscle fatigue estimator. Such an estimator would allow practitioners not only in sports but also in medical and research fields to assess muscle fatigue more effectively. Therefore, the present study aimed to (1) examine the validity of using RPE as an indirect indicator of muscle fatigue by investigating the relationship between RPE and muscle fatigue, as quantified by a new sEMG-based algorithm during explosive back squat (BS) exercises, and (2) compare the RPE-based muscle fatigue indicator with velocity loss to establish a more practical muscle fatigue marker that can be utilized in explosive resistance training programs.

## 2. Materials and Methods

### 2.1. Experimental Design

The present study aimed to evaluate the validity of using RPE as an indirect muscle fatigue marker and to compare it with velocity loss during explosive BS exercises. A counterbalanced crossover design was employed to minimize order effects. The explosive BS exercise was selected due to its widespread use in explosive resistance training programs. The experimental protocol consisted of two sessions, separated by at least 48 h. In the initial session, participants were given detailed instructions on the experimental procedures, and descriptive data were collected. An anchoring trial was then conducted to calibrate the maximum and minimum boundaries of perceived exertion for the subsequent experimental session [35,38]. The anchoring trial involved performing a single set of explosive BSs until physical failure. The number of successful repetitions was recorded to determine the target repetition count for the experimental conditions in the next session. During the experimental session, participants performed three sets of explosive BSs corresponding to 30% (L), 60% (M), and 90% (H) of the successful repetition number established in the anchoring trial. The L, M, and H conditions were implemented in a counterbalanced order. sEMG, Borg’s CR-10 score, and movement velocity were recorded for all three conditions and used for subsequent analysis.

### 2.2. Participants

The sample size was determined using statistical power analysis software (G*Power 3.1, Bonn University, Bonn, Germany). An ANOVA model with fixed effects, main effects, and interaction analyses was used, with input parameters set to an effect size of 0.4, an alpha level of 0.05, and a statistical power of 0.95 [16,38]. The minimum required number of participants was calculated to be 14. Consequently, 17 trained male participants were recruited for the present study, exceeding the minimum requirement for achieving adequate statistical power. The participants were active or former athletes, all of whom were engaged in daily resistance training. All participants were recruited from various sports teams and associated stakeholders. On average, the participants had 10.9 ± 3.5 years of competition experience and 4.5 ± 2.4 years of experience in resistance training specifically. All participants were proficient in resistance training techniques and had no history of neuromuscular or skeletal muscle injuries, nor were they undergoing any medical treatments at the time of the study. The descriptive data of the participants are as follows (mean ± standard deviation): age, 21.4 ± 2.6 years; body mass, 72.6 ± 11.9 kg; height, 172.8 ± 5.4 cm; body fat percentage, 16.0 ± 3.9%; and 1RM for the back squat, 130.0 ± 33.2 kg. All participants were thoroughly informed about the experimental procedures, measurements to be taken, potential risks, discomforts, and benefits of the study. Written informed consent was obtained from all participants prior to the experimental measurements. The study was conducted in accordance with the ethical guidelines outlined in the Declaration of Helsinki and was approved by the Human Ethics Committee of Waseda University (Approval No. 2023-112).

### 2.3. Initial Session

Prior to the start of the session, participants received detailed instructions on the purpose of the study, the items to be measured, and the potential risks and benefits. After providing this information, written informed consent was obtained from all participants. Descriptive characteristics of the participants were then measured using a bioelectrical impedance device (InBody 970, InBody Co., Ltd., Seoul, Republic of Korea). After the initial measurements, participants warmed up in preparation for the 1RM strength test. The warm-up protocol included five minutes of jogging, specific static stretching, dynamic stretching, and two sets of explosive BSs with 20 kg and 30 kg, performing six and eight repetitions, respectively. This warm-up protocol was adapted from previous studies and has been shown to enhance explosive performance [3,41]. Following the warm-up, participants’ 1RM for the BS was assessed according to standard strength training guidelines, using a standard Olympic barbell [28]. The 1RM measurement involved progressively increasing the weight until the participant successfully completed a parallel BS with the maximum load they could lift for one repetition.

After the initial measurements, participants were given a 5 min rest interval before proceeding to the next phase of the protocol. During this rest period, participants were instructed on how to use Borg’s CR-10 scale. This scale ranges from “nothing at all” (0 points) to “extremely strong” (10 points) and includes verbal descriptors for the perception of exertion [30]. Perceived exertion was defined as the subjective feelings of effort, discomfort, and fatigue in the lower body during the BS trials. Following the instructions on using the scale, participants were introduced to the lifting cadence. The cadence was set to a 2 s lowering phase (eccentric) followed by an explosive pushing phase (concentric) [38]. The cadence was controlled using a smartphone metronome application set to emit one beep per second. To ensure consistency in BS depth across trials, a pair of timing gates was placed at the height of a parallel squat (TCi Timing System, Brower Timing System LLC, Draper, UT, USA). When participants reached this depth, the sensors emitted a sound, signaling them to stand up explosively. Participants then performed a practice session using only the barbell to familiarize themselves with the cadence and lifting technique. After the practice session, an anchoring trial was conducted to establish the range of RPE values for the upcoming experimental session. Participants performed a single set of explosive BSs until they reached physical failure. The lower anchor for RPE, corresponding to a score of 0, was established when the participant sat in a relaxed state, indicating “nothing at all” [26,42]. The upper anchor, corresponding to a score of 10, was defined as the point at which the participant reached failure, representing a feeling of “extremely strong” exertion [26,38]. The weight used for the anchoring trial was set at 65% of the participant’s 1RM, consistent with the experimental conditions in the subsequent session [31,38]. A repetition was defined as lowering to the parallel position (timing gate activation) and then standing back up to the starting position.

### 2.4. Experimental Session

Before the start of the experimental session, all participants completed the same warm-up routine as in the initial session. After completing the warm-up, participants were instructed to perform an RPE reporting practice. Borg’s CR-10 scale was placed directly in front of the participants, making it easily visible and readable. The lifting cadence was set to 2 s for the lowering phase (eccentric) and an explosive raising phase (concentric), with a 2 s pause between repetitions. A metronome, set to beep once per second, was used to control the cadence. During the 2 s pause between repetitions, participants reported their RPE score for the most recent repetition using the Borg’s CR-10 scale. They were instructed to base their RPE ratings on the subjective exertion range established during the anchoring trial. If participants experienced a level of exertion that exceeded the high anchor (10 on the Borg’s CR-10 scale, corresponding to “extremely strong”), they were allowed to report scores above 10 to best describe their subjective feelings [30,43].

After completing the warm-up and RPE reporting practice, the L, M, and H conditions were administered to participants in a counterbalanced order. The required number of repetitions for each condition was determined based on the number of repetitions completed during the anchoring trial. To minimize the influence of the central governor effect (anticipatory responses to a given task), participants were not informed of the total number of repetitions until the second-to-last repetition [44,45]. They were instructed to stop immediately after completing the final repetition. Participants were asked to report their RPE score after each repetition by selecting a number during the 2 s pause. As the lifting progressed, participants might experience severe breathlessness and have difficulty maintaining the specified cadence. In such cases, they were encouraged to follow the metronome as closely as possible and focus solely on the subjective exertion experienced in their lower body. After completing each condition, participants were asked to report an overall RPE score for the most recent trial, reflecting their overall subjective exertion. A 5 min rest interval was provided between the experimental conditions.

### 2.5. Surface Electromyography

sEMG signals were measured from the vastus lateralis and vastus medialis muscles of the dominant leg. These two muscles were selected because they are known to exhibit similar activation patterns, allowing them to be averaged during multi-joint resistance exercises [46,47]. Bipolar sEMG electrodes were used for electromyographic recording. Two Ag/AgCl surface electrodes (ADMEDEC Co., Ltd., Tokyo, Japan) were placed on each muscle, with a 1 cm inter-electrode distance. Specifically, for the vastus medialis, electrodes were placed at 80% of the line between the anterior spina iliaca superior and the joint space in front of the anterior border of the medial ligament. For the vastus lateralis, electrodes were positioned at two-thirds along the line from the anterior spina iliaca superior to the lateral side of the patella [48]. Prior to electrode placement, the skin was shaved, abraded with sandpaper, and cleaned with alcohol swabs to reduce skin impedance. Electrodes were positioned according to anatomical landmarks and established guidelines for electrode placement [48,49]. The sEMG signals were sampled at a frequency of 1000 Hz (MARQ MQ-8, Kissei-Com Tech, Nagano, Japan) and were then amplified and transmitted to a dedicated computer. A high-speed camera (Grasshopper GRAS-03K2C, FLIR Systems Inc., Wilsonville, OR, USA) was used to record high-speed video synchronized with the sEMG signals. Raw sEMG data were recorded using specialized software (Vital Recorder Ver2.7.6.1611, Kissei-Com Tech, Nagano, Japan). Subsequently, the raw sEMG signals were segmented into individual repetitions and exported for further analysis. Only the concentric phases of each repetition were used for muscle fatigue quantification.

A fourth-order Butterworth bandpass filter (20–450 Hz) was applied to the sEMG signals to filter out noise. The filtered signals were then used to quantify muscle fatigue levels for each experimental condition. As discussed in the Introduction, the power spectral density of sEMG signals can be significantly affected during dynamic muscle contractions. To minimize these effects, a specialized analysis technique known as the spectral fatigue index (SFI) was used for muscle fatigue quantification in this study [13,14]. SFI has been shown to provide greater precision than traditional sEMG-based fatigue parameters, such as median and mean frequency, during dynamic contractions [10,13]. A fast Fourier transformation was applied to the sEMG signals to obtain their power spectrum. Spectral moments were used to extract characteristics of the power spectral density function and were calculated using Equation (1):(1)$$M_{k}=\int_{f_{min}}^{f_{max}}f^{k}\cdot PS\left(f\right)\cdot df$$
where *M_k_* is the spectral moment of order *k*, *PS*(*f*) represents the power spectrum at frequency *f*, and *f_min_* and *f_max_* correspond to the lower and upper cut-off frequencies of the bandpass filter, respectively. The SFI value was calculated as the ratio between the spectral moments of orders −1 and 5, as shown in Equation (2):(2)$$SFI=\frac{\int_{f_{min}}^{f_{max}}f^{-1}\cdot PS\left(f\right)\cdot df}{\int_{f_{min}}^{f_{max}}f^{5}\cdot PS\left(f\right)\cdot df}$$

The SFI was calculated for each repetition, and the relative changes in SFI values across different repetitions were determined by comparing them with the first repetition of each experimental condition. The SFI results from the vastus lateralis and vastus medialis muscles were averaged, as their electromyographic responses have been reported to be similar [46]. All SFI calculations were performed using Matlab (Matlab 2024a, Mathworks, Natick, MA, USA).

### 2.6. Velocity Loss

The barbell velocity during the BS was calculated using high-speed video (120 frames per second) recorded from the sagittal plane. Linear position transducers, such as GymAware, were not used in the current study because these devices could not be synchronized with the sEMG signals. The camera was positioned approximately 5 m away from the squat rack, facing the participant’s right side. A marker was placed at the end of the barbell to track the barbell’s trajectory. Pixel-to-distance calibration was conducted using a 1 m calibration stick positioned near the barbell’s edge, providing a reference for converting pixel measurements into real-world distances [50]. The barbell trajectory coordinates were automatically tracked using dedicated software (KineAnalyzer Ver4.0.2.1407, Kissei-Com Tech, Nagano, Japan). The concentric velocity was then calculated based on the barbell’s trajectory coordinates. These concentric velocities were separated into individual repetitions and exported for velocity loss calculation. The data extraction process was performed using customized Matlab code (Matlab 2024a, Mathworks, Natick, MA, USA). Intra-set velocity loss was determined by comparing the velocity of each repetition to the fastest repetition within each condition and was calculated on a repetition-by-repetition basis [3].

### 2.7. Statistical Analysis

The Shapiro–Wilk test was used to assess the normality of overall RPE, average velocity loss, and average SFI for each condition individually. The average SFI was found to be normally distributed, so a one-way ANOVA was applied, followed by a post hoc test with Bonferroni correction for multiple comparisons. In contrast, the overall RPE for the M condition and the velocity loss for the L and H conditions were not normally distributed. Therefore, Friedman’s test was used to evaluate differences in overall RPE and average velocity loss among the experimental conditions, with post hoc comparisons conducted using the Bonferroni correction.

To investigate RPE, SFI, and velocity loss during the BS exercise for the L, M, and H conditions, data from specific repetitions within each experimental condition were analyzed. Since the required number of repetitions varied for each condition and participant (L: 5.4 ± 1.0 repetitions; M: 10.4 ± 2.2 repetitions; H: 15.8 ± 3.2 repetitions), the first, median, and last repetitions were selected for comparison. A two-way ANOVA (3 conditions × 3 repetitions) was conducted to test for main and interaction effects of conditions and repetitions. If significant effects were found, post hoc comparisons were performed using Bonferroni correction for multiple comparisons.

For the correlation analysis, intra-set RPE, velocity loss, and SFI data were used. Spearman’s rho was calculated because some data were not normally distributed, as described in the previous paragraph. The correlation coefficients were categorized as follows: weak positive monotonic correlation for 0 < *p* ≤ 0.4, moderate positive monotonic correlation for 0.4 < *p* ≤ 0.8, and strong positive monotonic correlation for 0.8 < *p* < 1, based on an online recommendation from “Strength of Correlation” https://www.ncl.ac.uk/webtemplate/ask-assets/external/maths-resources/statistics/regression-and-correlation/strength-of-correlation.html (accessed on 8 November 2024). Since half of the data was collected within the 0–60% range of the maximum repetition number, only the non-overlapping portions of the three conditions were included in the correlation analysis [38,51]. Specifically, data from the 0–30% range of the maximum repetitions for the L condition, the 30–60% range for the M condition, and the 60–90% range for the H condition were analyzed. The correlation coefficients between SFI and RPE, as well as between SFI and velocity loss, were compared using Fisher’s r-to-z transformation and paired two-tailed tests. Statistical analysis was performed using SPSS 29.0 (SPSS Inc., Armonk, NY, USA), with the significance level set at *p* < 0.05.

## 3. Results

Significant effects in overall RPE were observed across the experimental conditions (*χ*^2^ = 33.522, *p* < 0.001, Kendall’s W = 0.986), with specific differences noted in the L vs. M (*p* = 0.008), M vs. H (*p* = 0.018), and L vs. H (*p* < 0.001) comparisons (Figure 1a). For average SFI, no significant effect was found for condition (*p* = 0.076, partial *η*^2^ = 0.102) (Figure 1b). Finally, significant effects in average velocity loss were observed across the experimental conditions (*χ*^2^ = 13.176, *p* < 0.001, Kendall’s W = 0.388), with specific differences between the L vs. M (*p* = 0.018) and L vs. H (*p* = 0.002) comparisons (Figure 1c).

Intra-set data were analyzed using two-way ANOVA, with the first, mid-point, and last repetitions extracted for this analysis. Significant interaction effects between conditions and repetitions were observed for RPE (*p* < 0.001, *F* = 60.639, partial *η*^2^ = 0.791). In the pairwise comparisons, RPE increased significantly from the first to the last repetition in all conditions (all *p* < 0.001), and significant differences between the conditions were observed at the mid-point (*p* < 0.001, *F* = 52.173, partial *η*^2^ = 0.765) and last repetition (*p* < 0.001, *F* = 82.219, partial *η*^2^ = 0.837) (Figure 2a). For SFI, significant main effects of condition (*p* = 0.035, *F* = 3.719, partial *η*^2^ = 0.189) and repetition (*p* < 0.001, *F* = 52.221, partial *η*^2^ = 0.765) were observed. Pairwise comparisons showed a significant effect of repetitions (*p* < 0.001, *F* = 31.197, partial *η*^2^ = 0.806); however, no significant differences between conditions were found for SFI. Additionally, no significant interaction between conditions and repetitions was observed for SFI (*p* = 0.052, *F* = 2.482, partial *η*^2^ = 0.134) (Figure 2b). Significant effects of conditions were observed for velocity loss (*p* < 0.001, *F* = 8.860, partial *η*^2^ = 0.356); however, no significant changes in velocity loss across repetitions were detected (Figure 2c).

The results of the Spearman correlation analysis are presented in Figure 3. A total of 269 BS repetitions were included in the analysis. Significant correlations were found between SFI and RPE (*r* = 0.325, *p* < 0.001) (Figure 3a) and between SFI and velocity loss (*r* = 0.224, *p* < 0.001) (Figure 3b). No significant difference was observed between the SFI–RPE and SFI–velocity loss correlations (*z* = 1.254, *p* = 0.210).

## 4. Discussion

As resistance exercise has grown in popularity, muscle fatigue—a common outcome—has become a significant concern due to its potential to increase the risk of injuries by impairing posture and movement control. Accurate quantification of muscle fatigue is crucial for ensuring safety, yet real-time assessment remains challenging, highlighting the need for a simple, effective tool. The RPE scale, which captures sensations of tiredness and discomfort, offers a practical way to quantify muscle fatigue due to its ease of use and its ability to reflect physiological responses. The purpose of this study was twofold: (1) to examine the validity of using RPE as a muscle fatigue marker by analyzing the relationship between RPE and muscle fatigue, quantified using a new sEMG-based algorithm during explosive BS exercise, and (2) to compare RPE with velocity loss to identify a more practical and versatile muscle fatigue indicator for use in explosive resistance exercise programs. The key findings of this study are as follows: (1) Both overall RPE and average velocity loss significantly increased with the prescribed number of repetitions, indicating that a higher prescribed training volume results in higher overall RPE and a more pronounced decrease in velocity during explosive BS exercise. (2) Significant increases were observed in both RPE and SFI as the lifting progressed, suggesting that RPE can serve as a real-time muscle fatigue marker during explosive BS exercise. However, (3) significant velocity loss increases were not observed across all experimental conditions, suggesting that velocity loss may not adequately reflect muscle fatigue when performing explosive BS exercises at moderate relative intensity (e.g., 65% 1RM), even when participants approached physical failure. (4) Significant but weak correlations between RPE and SFI, as well as between velocity loss and SFI, were found, indicating that the mechanisms of muscle fatigue in the leg muscles are complex due to specific physiological characteristics. Therefore, practitioners should consider a more comprehensive approach to quantifying muscle fatigue during explosive BS exercise.

Subjective exertion is related to a range of physiological responses, from central to peripheral, including metabolic and neuromuscular reactions induced by resistance exercises [35,52]. For example, RPE, blood lactate levels, and muscle activation have been shown to increase with higher relative intensity in resistance exercise [35]. These findings suggest that RPE is a comprehensive, indirect marker of the physiological responses triggered by resistance exercises. In this study, relative intensity was kept constant across all experimental conditions to eliminate the influence of weight, meaning that the significant differences in RPE observed may be attributed to the variations in the prescribed number of repetitions for the L, M, and H conditions. This result aligns with previous studies, which have shown that resistance exercise programs emphasizing muscular hypertrophy (with a higher prescribed volume) result in more pronounced metabolic, endocrine, and perceptual responses, even when performed at lower relative intensities [17,53]. Consistent with these findings, the H condition in this study, which had the highest number of repetitions and thus the highest volume, led to accelerated intramuscular perturbation and disruption of homeostasis (e.g., accumulation of hydrogen ions) [54]. Group III/IV muscle afferents, which generate feedback from local muscles, can cause significant increases in perceived exertion and subjective discomfort as a result of these physiological changes [47,55,56]. Moreover, disruptions in homeostasis affect muscle fiber conductivity, which can further influence the power spectral characteristics of the sEMG signal [13,14,57]. Such significant changes in intramuscular homeostasis are reflected in the power spectral density [12,57]. As a result, the power spectrum shifts from high to low frequencies due to muscle fatigue-related mechanisms, leading to a significant increase in SFI as the lifting progresses [58]. Although no significant differences in muscle fatigue between conditions were observed, the *p*-values and effect sizes suggest corresponding increases in RPE and SFI in both one-way (*p* = 0.076, partial *η*^2^ = 0.102) and two-way ANOVA (*p* = 0.052, partial *η*^2^ = 0.134). While the sample size was determined based on statistical power analysis, specific muscle fatigue patterns, such as a higher proportion of fatigue-resistant muscle fibers and larger capacity to take up metabolic byproducts in the leg muscles, may have influenced the muscle fatigue levels induced by the experimental settings [59,60]. The synchronous increases in RPE and SFI suggest that as muscle fatigue levels rise, subjective discomfort, as reflected by RPE scores, increases correspondingly. Therefore, RPE may serve as a simplified muscle fatigue indicator during explosive BS exercises. Strength coaches and personal trainers can monitor RPE on a repetition-by-repetition basis or at predetermined points (e.g., the first, mid-point, and last repetition) to track muscle fatigue during the lifting process. Unlike velocity-based assessments, RPE-based muscle fatigue assessment does not require specialized devices or equipment (e.g., linear encoders), making it a more practical option for assessing muscle fatigue in explosive resistance exercise settings.

With the widespread use of velocity-based training, practitioners have increasingly adopted velocity-measuring devices (e.g., GymAware) to monitor fatigue levels during power and explosive performance-oriented resistance training programs. Previous studies have validated the relationship between velocity loss and fatigue-related markers in explosive resistance exercises, demonstrating significant associations between velocity loss and both metabolic and mechanical stress induced by fatigue [3,26]. In the present study, although a significant relationship was observed between SFI and velocity loss, the correlation coefficient was relatively weak (*r* = 0.224, *p* < 0.001), which appears to contradict previous findings. One potential reason for this discrepancy could be the training volume settings. In this study, the H condition was set at 90% of the repetition number that leads to physical failure. Although the relative intensity (65% 1RM) was consistent with previous studies (e.g., 12RM), multiple sets were performed for each condition [3]. Thus, a single set of explosive BSs may not have been sufficient to induce significant muscle fatigue in the leg muscles, even when the repetition number approached physical failure. Another reason could be attributed to the physiological characteristics of the leg muscles. Previous studies have indicated that approximately two-thirds of the vastus lateralis muscle is composed of type I (fatigue-resistant) muscle fibers [59]. Additionally, significant changes in blood ammonia concentration were observed only after high-repetition protocols (e.g., 10–12 repetitions for 3 sets at 12–10RM), while mechanical fatigue indicators (e.g., loss in impulsive velocity) did not show significant decreases [3]. In contrast, RPE increased significantly, which aligns with previous findings in explosive leg press exercises [26]. Similar results were observed in our previous research, suggesting a more complex fatigue pattern in the leg muscles compared with the upper-body muscles [39]. The results of the correlation analysis support this perspective, as a significant but relatively weak correlation was observed between RPE and SFI (*r* = 0.325, *p* < 0.001). A similar, but even weaker, relationship was found between velocity loss and SFI (*r* = 0.224, *p* < 0.001). These results suggest that fatigue patterns in the leg muscles during explosive BS exercises are more complex, making it difficult to indirectly predict muscle fatigue. The increase in RPE is likely driven by metabolic perturbations and related responses, such as breathlessness induced by an increase in repetition number.

Previous studies examining submaximal set configurations found significant correlations between the OMNI scale (a resistance exercise-specific RPE scale) and mean concentric velocity during leg press exercises [26]. For mechanical-based fatigue markers (e.g., velocity loss and force output), significant relationships have been examined between physiological measures and mechanical parameters [3,54]. While these studies demonstrated that RPE could be used as a fatigue indicator, direct evidence linking RPE to muscle fatigue has been limited. Our study examined RPE, velocity loss, and muscle fatigue together, aiming to bridge the gap between these three variables; however, our findings only partially align with previous recommendations. Both RPE and velocity loss can be used as simplified muscle fatigue indicators, but velocity loss is sufficient to serve as an independent muscle fatigue marker. Coaches and trainers could average velocity loss across a specific set to obtain an estimate of average fatigue level, allowing them to visualize muscle fatigue for a given training session. However, velocity loss should not be used as a muscle fatigue indicator on a repetition-by-repetition basis, as velocity may not decrease even when significant muscle fatigue has already developed. Therefore, practitioners should assess muscle fatigue with caution and consider using a more comprehensive approach, incorporating multiple fatigue-related mechanical measures when applying velocity loss as fatigue indicator during explosive BS exercise.

The present study resolves an important blind spot for the application of velocity-based training. With the increasing popularity of velocity-based training worldwide, numerous studies have begun to advocate for its validity and practical application, and many low-cost commercial encoders are now available for use in resistance training scenarios [3,25,26]. Most studies investigating the effectiveness of velocity-based training have primarily focused on its impact on athletic performance and have evaluated only performance-related parameters (e.g., power and velocity) through velocity-based feedback. As a result, many researchers have started to suggest using velocity-related parameters to predict other performance metrics, such as force and fatigue, in sports-related scenarios [3,26,61]. For example, Mayo et al. indicated that RPE could be used as a fatigue indicator alongside velocity loss; however, direct fatigue-related measures were not assessed in their study [26]. While the positive impact of velocity-based training on athletic performance is well documented, there remains a lack of sufficient physiological evidence supporting its broader applications. In the present study, muscle fatigue was observed even though there was no significant velocity loss, despite using a highly sensitive muscle fatigue quantification algorithm (Figure 2b,c) [14]. This finding underscores the need for coaches and trainers to exercise caution when using velocity-based predictions for physiological parameters. In addition to the lack of physiological evidence for velocity-based training, inconsistencies in velocity-assessing devices may also significantly influence training outcomes [24,25,62]. For instance, some studies have reported that certain velocity/position transducers exhibit limited precision and cannot be used interchangeably when performing velocity-based training [24,62]. In such cases, relying solely on velocity assessment could result in an oversight of negative training outcomes and even lead to long-term negative effects, such as overtraining. Consequently, practitioners should not rely exclusively on velocity transducers or velocity-based predictions of physiological responses. Velocity-based training is a relatively new methodology, and its impact on physiological responses has not yet been fully substantiated. Coaches and trainers should exercise caution when using velocity-based training, especially in lower-body resistance exercises. It is recommended to use velocity transducers with the highest precision available and to combine velocity measurements with other physiological-related assessments to minimize deviations and potential negative effects associated with velocity-based training.

The present study has several limitations. First, to obtain intra-set RPE and minimize the influence of bottom rebound, a predetermined lowering and pause tempo was used during the lifting process. It is challenging to completely avoid the influence of lifting cadence on RPE and fatigue responses, as participants may have difficulty maintaining the cadence during the latter half of the lifting process. Participants may also have had difficulty isolating the influence of breathlessness. Although we did not assess breathlessness in the present study, some participants reported that it affected their subjective feelings. Second, this study assessed only the explosive BS exercise, as it is a key exercise for developing lower-body strength and power. However, these findings may not be directly applicable to upper-body resistance exercises. Previous research suggests that muscle fatigue in the lower body is more complex due to physiological differences such as lactate kinetics and muscle fiber composition [39,60]. Third, although the sample size was determined using statistical power analysis, it remains relatively small, which may have limited the statistical significance of some analyses. For example, in the SFI analysis, significant differences were not observed in the ANOVA results (one-way, *p* = 0.076; two-way condition: *p* = 0.069; interaction: *p* = 0.052). Lastly, we recruited only male participants in the present study, which may limit insight into sex-specific responses. Future studies should aim to include a larger sample size to ensure sufficient statistical power, reduce the potential influence of lifting tempo on intra-set data, quantify breathlessness as supporting evidence, examine sex-specific characteristics, and directly compare differences between upper- and lower-body explosive exercises.

## 5. Conclusions

The present study demonstrated that overall RPE, average velocity loss, and average SFI changed correspondingly, indicating that both RPE and velocity loss could serve as global indicators of muscle fatigue for a given set or a single training session of BSs. For intra-set data, RPE and SFI changed in a similar manner, suggesting that RPE could also be used as a simplified real-time muscle fatigue indicator during the lifting process. However, velocity loss did not show corresponding changes with SFI, suggesting that velocity loss may be inappropriate for indirect muscle fatigue assessment in speed and power-focused lower-body exercises such as explosive BSs. Significant but weak relationships were observed between SFI and RPE as well as between SFI and velocity loss. This indicates that the underlying mechanisms of fatigue in the leg muscles are more complex, likely due to specific characteristics such as muscle fiber composition and related metabolic kinetics. Given the ease of use of RPE, it is a valid and practical indicator of muscle fatigue in explosive lower-body exercises. Coaches and personal trainers can rely on RPE to assess muscle fatigue when direct measurement of fatigue is not feasible. However, with the increasing use of velocity-based training to enhance athletic performance, the physiological evidence supporting its use remains insufficient. It is important to note that using velocity-based assessments alone may lead to an oversight of negative training outcomes, including muscle fatigue. Practitioners should aim to use velocity transducers with the highest possible precision and/or combine them with other physiological measures to reduce biases in velocity-based indirect physiological assessments, especially in lower-body exercises. Therefore, practitioners should exercise caution when using velocity alone for muscle fatigue assessment in explosive lifting programs targeting the lower body.

## Figures and Tables

**Figure 1 jfmk-09-00238-f001:**
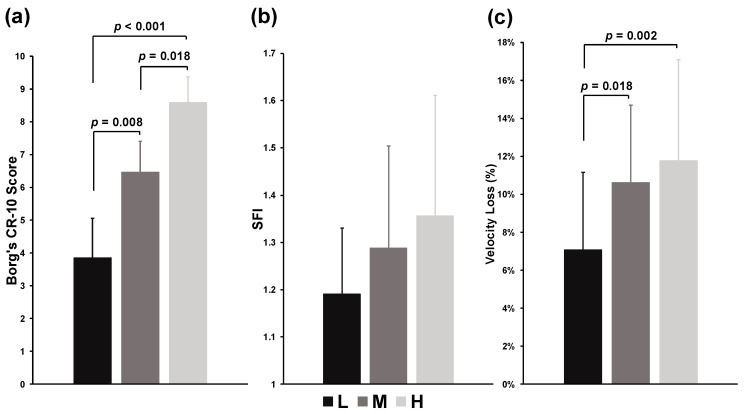
Overall ratings of perceived exertion (RPE) (**a**), average spectral fatigue index (SFI) (**b**), and average velocity loss (**c**) of back squat tasks. The *p* value represents the differences and significance levels between the experimental conditions.

**Figure 2 jfmk-09-00238-f002:**
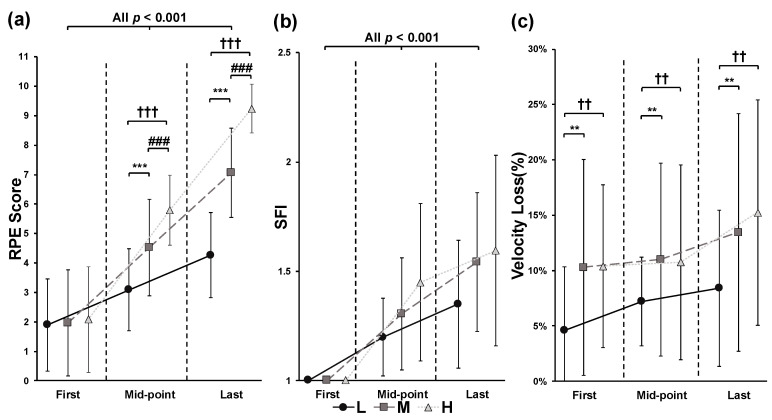
Rating of perceived exertion (RPE) (**a**), spectral fatigue index (SFI) (**b**), and velocity loss (**c**) during low (L, circle with solid lines), medium (M, square with dashed lines), and high (H, triangle with dotted lines) volume condition of exercises. Asterisk represents a significant difference as L compared with M condition, ** *p* < 0.01, *** *p* < 0.001; Hash represents a significant difference as M compared with H condition, ### *p* < 0.001; Dagger represents a significant difference as L compared with H condition, †† *p* < 0.05, ††† *p* < 0.001. The *p* value indicates the differences and significance levels between repetitions.

**Figure 3 jfmk-09-00238-f003:**
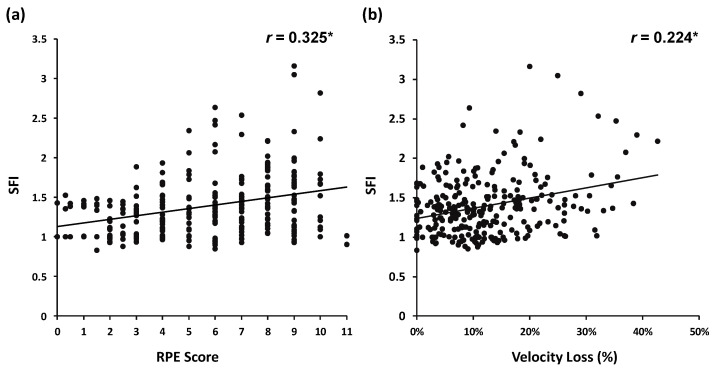
Spearman’s rho of spectral fatigue index (SFI) ratings of perceived exertion (RPE) scores (**a**), and SFI–velocity loss (**b**), during back squat tasks. * represents the significant level of correlation coefficients, *p* < 0.001.

## Data Availability

The data presented in this study are available on request from the corresponding author. The data are not publicly available due to high-speed video was recorded in this study, which may exposures the face of participants.

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
