# Peer review of "Can Perceived Exertion and Velocity Loss Serve as Indirect Indicators of Muscle Fatigue During Explosive Back Squat Exercise?"

_jfmk, 2024, doi:10.3390/jfmk9040238_

Round 1
Reviewer 1 Report
Comments and Suggestions for Authors
The subject matter of this article is current and has great applicability.
Introduction
There is good initial argumentation about fatigue based on EMG, in which I could have added the calculation of fatigue through the median of the signal.
It is very positive to indicate the possibility of a new algorithm for sEMG, although the authors indicate that this type of device is very expensive and there is some contradiction.
It is indeed expensive, so they could adhere to recent studies that link the RPE to explosive tasks, such as CMJ jumps on a contact platform, a more accessible device that provides objective data to improve reproducibility.
Methods
In the description, the number of sets performed for each subject should be added until the 1RM was reached.
Electrode positioning was performed only on the medial and lateral vastus muscles. What is the reason for not placing it on the gluteus or rectus femoris muscles? It would provide more information in polyarticular exercises such as this one.
I missed the accompaniment of some images.
The loss of velocity of the bar is not simple, and is more precise when an encoder is placed. This was indirect support.
Discussion
The reflection is very interesting; perhaps one could add what would be the load from which the perception of effort rises to the highest level to make it more accessible to the reader.
In the limitations, it could be added that it has not been conducted with a female sample.
Reviewer 2 Report
Comments and Suggestions for Authors
This manuscript is well-organized and written. The introduction suggested a gap in the literature that needed to be examined and concluded with a clear purpose statement. Generally, the methods section is comprehensive, although some additional information is needed concerning the description of EMG and statistical analysis. Results, discussion, and conclusion clear and comprehensive and related to the study's primary purpose. There were a few minor word choice errors. See PDF for detailed comments.

Reviewer 3 Report
Comments and Suggestions for Authors
Review jfmk-3319849-peer-review-v1
The paper Can Perceived Exertion and Velocity Loss Serve as Indirect Indicators of Muscle Fatigue During Explosive Back Squat Exercise? examines the validity of perceived exertion (RPE) and velocity loss as markers of muscle fatigue in explosive back squat exercises. It explores non-invasive methods for monitoring fatigue, which is important for enhancing performance and preventing injury. The authors address a research gap regarding RPE and velocity loss in explosive movements, highlighting the study's importance. The article is well-cited, mostly from the past decade, and effectively contextualizes its findings within recent research on muscle fatigue, RPE, and velocity-based training.
The methodology is robust, featuring a counterbalanced crossover design and detailed data collection protocols. With 17 participants, the sample size is statistically sufficient, enhancing the study's reliability, and using both RPE and SFI as fatigue metrics adds depth to the analysis. However, focusing solely on the back squat may limit the findings' applicability to other explosive exercises (please, add this to the study limitation section). While standardized equipment and protocols ensure reproducibility, the high equipment requirements could restrict applicability in general sports settings without access to such tools.
The results section references figures that are not included in the paper, making it impossible to assess them.
The conclusions are clear and supported by the results. The authors suggest that while RPE is a useful fatigue marker, velocity loss should not be used alone in explosive lower-body exercises due to its weak correlation with muscle fatigue. The study's practical insights offer valuable guidance for coaches. It acknowledges limitations, such as a small sample size and focus on a single exercise, and suggests future research could explore other exercises or a larger, more diverse group. The recommendation to combine RPE with other indicators aligns with the findings and could be explored with additional non-invasive metrics.
In conclusion, Zhao et al.’s paper is a well-executed and relevant study for the field of sports science. It successfully addresses a critical gap in muscle fatigue monitoring methods for explosive exercises.
Three points for improvement:
1. The study only uses trained male participants, which may limit generalizability. A note on this limitation, particularly in applying findings to other populations, would add transparency.
2. The detailed descriptions of electrode placement, sEMG signal filtering, and other technical steps, while scientifically rigorous, might benefit from simplification or summarization for reader accessibility.
3. Please add figures in the results section.
